# Improving Shape-Sensing Robotic-Assisted Bronchoscopy Outcomes with Mobile Cone-Beam Computed Tomography Guidance

**DOI:** 10.3390/diagnostics14171955

**Published:** 2024-09-04

**Authors:** Sami I. Bashour, Asad Khan, Juhee Song, Gouthami Chintalapani, Gerhard Kleinszig, Bruce F. Sabath, Julie Lin, Horiana B. Grosu, Carlos A. Jimenez, Georgie A. Eapen, David E. Ost, Mona Sarkiss, Roberto F. Casal

**Affiliations:** 1Department of Pulmonary and Critical Care Medicine, Michael E. DeBakey VA Medical Center, Houston, TX 77030, USA; sami.bashour@va.gov; 2Department of Pulmonary and Critical Care Medicine, Ochsner Health Rush, Meridian, MS 39301, USA; 3Department of Biostatistics, The University of Texas MD Anderson Cancer Center, Houston, TX 77030, USA; 4Siemens Medical Solutions USA Inc., Malvern 19355, PA, USA; 5Siemens Healthcare GmbH, 91301 Forchheim, Germany; 6Department of Pulmonary Medicine, The University of Texas MD Anderson Cancer Center, Houston, TX 77030, USA; 7Department of Anesthesia and Peri-Operative Medicine, The University of Texas MD Anderson Cancer Center, Houston, TX 77030, USA

**Keywords:** robotic bronchoscopy, cone-beam computed tomography, lung nodules

## Abstract

Background: Computed tomography to body divergence (CTBD) is one of the main barriers to bronchoscopic techniques for the diagnosis of peripherally located lung nodules. Cone-beam CT (CBCT) guidance is being rapidly adopted to correct for this phenomenon and to potentially increase diagnostic outcomes. In this trial, we hypothesized that the addition of mobile CBCT (m-CBCT) could improve the rate of tool in lesion (TIL) and the diagnostic yield of shape-sensing robotic-assisted bronchoscopy (SS-RAB). Methods: This was a prospective, single-arm study, which enrolled patients with peripheral lung nodules of 1–3 cm and compared the rate of TIL and the diagnostic yield of SS-RAB alone and combined with mCBCT. Results: A total of 67 subjects were enrolled, the median nodule size was 1.7 cm (range, 0.9–3 cm). TIL was achieved in 23 patients (34.3%) with SS-RAB alone, and 66 patients (98.6%) with the addition of mCBCT (*p* < 0.0001). The diagnostic yield of SS-RAB alone was 29.9% (95% CI, 29.3–42.3%) and it was 86.6% (95% CI, 76–93.7%) with the addition of mCBCT (*p* < 0.0001). There were no pneumothoraxes or any bronchoscopy-related complications, and the median total dose–area product (DAP) was 50.5 Gy-cm^2^. Conclusions: The addition of mCBCT guidance to SS-RAB allows bronchoscopists to compensate for CTBD, leading to an increase in TIL and diagnostic yield, with acceptable radiation exposure.

## 1. Introduction

The recommendation of low-dose computed tomography (CT) for lung cancer screening by most society guidelines and the widespread use of CT in clinical practice has left physicians with an enormous challenge: the diagnostic evaluation of peripheral lung nodules [1,2]. Since most of these lung nodules are found to be benign, biopsy is often necessary to establish a diagnosis. Different modalities can be utilized to obtain a biopsy of these nodules: bronchoscopy, CT-guided transthoracic needle biopsy (CT-TTNA), and video-assisted thoracoscopic surgery (VATS) [3]. The use of bronchoscopy is steadily increasing because it is a relatively safe procedure, and it can also provide mediastinal staging along with diagnosis. Nevertheless, ultrathin bronchoscopy with radial-probe endobronchial ultrasound (RP-EBUS), virtual bronchoscopy navigation, and different platforms for electromagnetic navigation have all failed to surpass a diagnostic yield of 70% in properly designed prospective trials [4,5,6,7,8]. While there has been increasing enthusiasm with the advent of robotic bronchoscopy, as evident in a recent meta-analysis by Zhang and coworkers, data are still derived from small prospective studies and retrospective studies with significant heterogeneity in the definition of diagnostic yield, making it difficult to draw any strong conclusions [9].

The recent addition of cone-beam CT (CBCT) (“fixed” and “mobile” platforms) to the bronchoscopists’ armamentarium has highlighted the phenomena of intra-operative atelectasis and CT-to-body divergence (CTBD) as some of the barriers to obtaining diagnostic samples with the above-mentioned navigational techniques [10,11,12,13,14]. Therefore, CBCT guidance to confirm “tool in lesion” (TIL) before obtaining samples could potentially improve the diagnostic yield of bronchoscopy for peripheral lung nodules. A few studies have shown the potential of “fixed” CBCT (fCBCT) to increase diagnostic and navigational yield for peripheral bronchoscopy [10,15,16,17,18]. Unfortunately, fCBCT equipment is bulky, costly, and typically attached to the floor or ceiling of interventional radiology suites or hybrid operating rooms which are only available to a very limited number of bronchoscopists. In the past few years, C-arms with 3D reconstruction capabilities have become available, and we will refer to them as “mobile” CBCT (mCBCT). Compared to fCBCT, these newer platforms have a smaller field-of-view, provide slightly lower quality images, and take 30–60 s for image acquisition. However, they do have several advantages such as their mobility, smaller footprint, and lower cost. This sparked our interest in investigating its added value to shape-sensing robotic-assisted bronchoscopy (SS-RAB) for the diagnosis of peripheral lung nodules. This specific combination of technologies (SS-RAB and mCBCT) has been reported in a small pilot prospective study by Reisenaur and coworkers, as well as in a retrospective series by Husta and coworkers, with promising results [13,19].

In the current clinical trial, we hypothesized that the addition of m-CBCT could improve the rate of TIL and the diagnostic yield of SS-RAB for peripherally located lung nodules.

## 2. Methods

### 2.1. Study Setting and Subjects

We conducted our study at the University of Texas MD Anderson Cancer Center with Institutional Review Board (2020-0760) approval. Enrollment started in October 2021 and ended in June 2023. Adult patients referred to pulmonary medicine for diagnosis of a peripheral lung nodule of 1 to 3 cm in diameter located in the outer two-thirds of the lungs were eligible. Patients with more than one target, contraindication for general anesthesia, pregnant or breast-feeding women, and those who could not provide informed consent were excluded from our study.

### 2.2. Study Design and Procedures

This was a prospective, single-arm study evaluating the impact of mCBCT guidance on the diagnostic yield and tool in lesion (TIL) rates of SS-RAB for peripheral lung nodules. See study procedures flowchart in Figure 1. Robotic bronchoscopy was performed following our standard of care with total intravenous anesthesia including neuromuscular blocking agents in all cases. Previously described ventilatory or positional strategies to prevent atelectasis were utilized when appropriate [20,21]. Nodal staging with endobronchial ultrasound (EBUS) was allowed, but only after robotic bronchoscopy. SS-RAB was performed with Ion Endoluminal Robotic Bronchoscopy System (Intuitive Surgical, Sunnyvale, CA, USA) and mCBCT was performed with Cios 3D Spin Mobile (Siemens Healthineers, Forchheim, Germany). The P4 software with “integration” of these two technologies (mCBCT spin transferred back to Ion platform to update the location of the target) was not available at the start of the study and hence it was not utilized in any study patient. mCBCT scans consisted of 30 s spins with normal or high resolution. Bronchoscopy started with an airway exam and clearance of secretions with a flexible bronchoscope (bronchoscope 1TH-190 Olympus America, Inc., Center Valley, PA, USA). This was followed by standard registration and navigation with SS-RAB towards the virtual target. RP-EBUS was not utilized. Once the virtual target was identified, the needle was deployed under 2D fluoroscopic guidance directed towards the center of the virtual target. The “near” and “far” distances provided by the navigational system were utilized to calculate the depth of the needle (and to ensure the needle would reach the center of the target). Once the needle was deployed, the first mCBCT spin was performed and it was independently evaluated by a physician (a CT-reading physician) other than the bronchoscopy operator. If TIL was confirmed, the bronchoscopy operator was allowed to proceed with sampling (transbronchial needle aspiration (TBNA) with 3 passes at a minimum), without being able to review the mCBCT images. Other additional tools could be used at the discretion of the operator. TIL or cytologic diagnosis obtained at this point were considered SS-RAB outcomes. If the virtual target was not reached, TIL was not seen with the first mCBCT spin, or cytologic diagnosis was not obtained despite TIL (by rapid on-site examination), the bronchoscopy operator was then allowed to review the mCBCT images, and based on those, maneuver with the robotic bronchoscope or needle to re-navigate, re-orient, re-deploy the needle, or utilize a different biopsy tool, and continue to attempt to achieve TIL and a diagnosis. The bronchoscopy operator was allowed to obtain up to 5 mCBCT spins in total for navigation, with any needed adjustments (robotic or tool) in between to obtain TIL or diagnosis (Figure 1). TIL or diagnosis obtained from the second to the fifth mCBCT spin were considered SS-RAB + mCBCT outcomes. 

### 2.3. Primary, Secondary Endpoints, and Definitions

The primary endpoint was to determine the added value of mCBCT to SS-RAB for the diagnosis of peripheral lung nodules. We utilized a strict definition of diagnostic yield in accordance with the recommendations from the American Thoracic Society (ATS) and the American College of Chest Physicians (ACCP) [8]. Secondary endpoints included TIL, sensitivity for malignancy, overall complications, and radiation exposure. The gold standard to calculate sensitivity for malignancy was either surgical pathology (from lung resection when available), CT-TTNA, or 12-month clinical and radiographic follow-up. All bronchoscopic samples showing malignancy were considered true positives. Cases where bronchoscopic samples were not diagnostic of malignancy but malignancy was later confirmed by either CT-guided TTNA, surgery, or radiographical progression of disease were considered false negatives. TIL was defined as the needle within lesion in all three axes—axial, sagittal and coronal. mCBCT images were evaluated in all three axes, and the shortest distance from the needle to the target (“tip” of the needle to “edge” of the target) observed in the first mCBCT spin was recorded for patients who did not have TIL (in whichever axis this shortest distance was found). Intra-bronchoscopy and post-bronchoscopy complications were defined a priori, collected during bronchoscopy, and extracted from medical records. The fluoroscopy time, number of mCBCT spins, and radiation exposure were recorded and reported following the guidelines of the International Commission on Radiological Protection [22]. Radiation exposure measured as the dose–area product (DAP) was defined as product of dose and beam area (Gy-cm^2^) and it was measured using an ionization chamber placed between the X-ray tube/collimator setup and the patient. Other relevant data collected included demographics, patient characteristics, target characteristics (anatomic location, distance to the pleura, size, radiographic characteristics, and the presence of a bronchus sign), and procedure characteristics (e.g., duration, which was defined as first scope “in” to last scope “out” and concomitant mediastinal staging).

### 2.4. Statistics

The sample size calculation was performed assuming a diagnostic yield of 60–70% for robotic bronchoscopy without mCBCT guidance. With the addition of mCBCT guidance we expected to reach diagnosis in 50% of those non-diagnostic subjects (non-diagnostic with SS-RAB alone). We calculated that a sample size of 67 subjects would yield approximately 20–27 non-diagnostic subjects, assuming a 30–40% non-diagnostic rate for SS-RAB. With 20 non-diagnostic subjects, it would achieve 80% power to detect a difference of 30% in diagnostic yield (comparing H0: 0.2 vs. H1: 0.5) using a two-sided binomial test with an alpha of 0.05 (PASS 2005, NCCS, Kaysville, UT, USA). This sample size would produce a 95% confidence interval equal to the sample proportion (diagnostic yield after adding mCBCT in non-diagnostic subjects) plus or minus 0.23 when the estimated proportion is 50%.

Descriptive statistics (mean (SD) or median (range), frequency (%)) were used to summarize the patient characteristics. Diagnostic yield, TIL, and sensitivity for malignancy for SS-RAB and addition of mCBCT along with their 95% CIs were estimated. mCBCT-added diagnostic yield, in a subgroup with non-diagnostic subjects by SS-RAB, was estimated along with its 95% CI. The two-sided exact binomial test was used to assess if diagnostic yield with the addition of mCBCT was significantly different from 0.5 in the subgroup with non-diagnostic subjects by SS-RAB. McNemar’s test was used to compare SS-RAB alone with the addition of mCBCT in terms of TIL and diagnostic yield. A p-value of less than 0.05 was utilized to indicate statistical significance. SAS 9.4 (SAS Institute Inc., Cary, NC, USA) was used for data analysis.

### 2.5. Study Registration

This study is registered in ClinicalTrials.gov as number NCT04995172.

## 3. Results

A total of 67 subjects were enrolled in the trial and underwent bronchoscopy. The baseline characteristics are summarized in Table 1. The median nodule size was 1.7 cm (range, 0.9–3 cm) and the median distance from the pleura was 0.7 cm (range, 0–4.7 cm). The bronchus sign was only present in 37.3% of the nodules. The procedure characteristics are summarized in Table 2. EBUS staging was performed in 47 (70%) of the cases and the median total procedure time was 51 min (range 21 to 127 min). TIL was achieved in 23 patients (34.3%) with SS-RAB alone, and 66 patients (98.6%) with the addition of mCBCT (*p* < 0.0001). Among the 44 patients who did not achieve TIL with SS-RAB alone, 43 (97.7%, 95% CI, 88.0–99.9%) achieved TIL with the addition of mCBCT. Diagnostic yield of SS-RAB alone was 29.9% (95% CI, 29.3–42.3%) and it was 86.6% (95% CI, 76–93.7%) with the addition of mCBCT (*p* < 0.0001). The addition of mCBCT helped us achieve a diagnosis in 38 of the 47 cases in which SS-RAB alone could not achieve a diagnosis (80.9%; 95% CI, 66.7–90.9%) (*p* < 0.0001). Sensitivity for malignancy of SS-RAB was 28.8% (95% CI, 17.8–42.1%) and specificity was 100% (95% CI, 63.1–100%). Sensitivity for malignancy of SS-RAB + mCBCT was 89.8% (95% CI, 79.2–96.2%) and specificity was 100% (95% CI, 63.1–100%). The median shortest distance from the needle to the target was 0.2 cm (interquartile range (IQR), 0.47 cm). Diagnoses are summarized in Table 3. Radiation data were available in 63 of the 67 patients and they are described in Table 4. There were no pneumothoraxes and no bronchoscopy-related complications. One patient developed atrial flutter during bronchoscopy, and one patient had hypotension requiring fluids during recovery. No other complications were recorded.

## 4. Discussion

To the best of our knowledge, this is the largest prospective trial with the combination of SS-RAB and mCBCT, and the only study comparing outcomes of SS-RAB alone and with the addition of mCBCT. Our trial demonstrated a substantial increase in TIL rates and diagnostic yield rendered by the addition of mCBCT guidance. The successful combination of these two technologies also resulted in safe procedures with no cases of pneumothorax or any other intraoperative complications and demonstrated acceptable radiation exposure metrics.

Our study has some similarities with a prior prospective pilot study conducted by Reisenaur and coworkers, which combined these same two technologies [13]. Their primary outcome was TIL, and their secondary outcomes included diagnostic yield, CTBD, and radiation exposure. The authors enrolled 30 patients and their population was comparable to ours with a high prevalence of malignancy (73%), nodules with a mean diameter of 1.7 cm, and a bronchus sign in 40% of the cases. In contrast with our trial, the authors defined TIL by the position of the robotic catheter (instead of the biopsy tool) with respect to the lesion; they did not report separate outcomes before and after the addition of mCBCT, and they utilized RP-EBUS. Their study showed a TIL rate of 100%, a diagnostic yield of 93.3%, and no cases of pneumothorax. The authors performed an extensive analysis of CTBD, calculating the displacement of the target between the pre-procedural and intra-op CT scans. They proposed two definitions of CTBD: 10% or less of overlap of the target between pre-procedure and intra-procedure scans, and > 1 cm divergence between the target centers. In total, 50% of the cases demonstrated divergence following the former definition, and 60% following the latter. Our data are congruent with these findings, with TIL only achieved in 34.3% of the cases when inserting the needle in the center of the virtual target (at the first mCBCT). The degree of CTBD was calculated differently in our study (shortest distance from needle to target during intra-op mCBCT), the median was 0.2 cm and the IQR of 0.47 cm. This was performed purposefully to investigate specifically how close the needle was from reaching the target when we did not achieve TIL, which is what we consider “clinically relevant” CTBD. For example, if we only measured target to target divergence (between pre- and intra-op CT), we could have a divergence of 1 cm in a 2 cm lesion, and still obtain TIL (and potentially a diagnosis), because there may still be overlap. It is truly important to emphasize that regardless of the degree of CTBD, the addition of mCBCT allowed the proceduralist to make the required corrections and obtain TIL in almost all cases in both studies.

A retrospective study from Husta and coworkers neatly evaluated the incremental contribution of mCBCT to SS-RAB [19]. Their primary outcome was TIL between the first and last mCBCT spin, and some of the secondary outcomes included the distance from the sampling tool to the center of the lesion and diagnostic yield utilizing an “intermediate” definition [23]. For their primary outcome group, the authors selected patients who underwent at least two mCBCT spins during SS-RAB. Unfortunately, of the 154 identified patients, they could only include 96 (102 targets), with 58 patients being excluded due to incomplete mCBCT imaging data. The nodule size was 1.58 cm (1.1–2.1 cm), a bronchus sign was present in 30%, 50% of the nodules were located in the inner two-thirds of the lungs, and they utilized RP-EBUS in 88% of the cases. On the first mCBCT spin, 48 lesions (47%) were classified as TIL, and this increased to 94% by the last mCBCT spin (*p* < 0.0001). The mean distance from the tip of the needle to the center of the target was 1.04 ± 0.51 cm. The diagnostic yield in the primary outcome group was 72% and there were no pneumothoraxes. Our trial overall supports the findings of this retrospective series which combined the exact same technologies, highlighting again the substantial incremental navigational and diagnostic value provided by the intra-op 3D imaging with mCBCT, as well as a potential reduction in the pneumothorax rate when compared with standard navigational techniques without 3D imaging guidance.

SS-RAB has also been prospectively studied in combination with fCBCT (Dyna CT, Siemens Medical Solutions, Malvern, PA, USA) by Benn and coworkers [24]. The authors did not report specific endpoints or their definition of diagnostic yield, but based on the diagnosis they described, we can assume they did not utilize a “strict” definition of diagnostic yield. Their study included 59 nodules in 52 patients. The median nodule size was 1.7 cm, 10 (17%) of the nodules were > 3 cm in largest dimension, and 9 (15%) were purely of ground-glass characteristics. They reported a diagnostic yield of 86%, a sensitivity for malignancy of 84% and two cases of pneumothorax, one requiring a chest tube.

In lieu of the rapid adoption of CBCT as an aid in navigation and sampling of peripheral lung lesions via bronchoscopy, it is of utmost importance for bronchoscopists to understand the principles of ionizing radiation and to be mindful of its exposure to patients, physicians, and personnel. A recent statement from the World Association of Bronchology and Interventional Pulmonology (WABIP) addresses this most relevant issue and provides practice guidelines to minimize radiation [25]. Moreover, this document recommends the standards for reporting radiation exposure in interventional pulmonology procedures, which are vital to facilitate comparisons between different systems, approaches, and outcomes. Following these recommendations, we reported the following metrics in the current trial: fluoroscopy time, DAP of fluoroscopy, number of CBCT spins, and DAP of CBCT. We also refrained from reporting the “effective dose” in Sieverts, since the conversion from DAP to effective dose requires a large number of assumptions (depending on the imaging system, field of view, imaging protocol, etc.) and can lead to inaccurate comparisons. The radiation exposure metrics described in our trial are in alignment with the previously reported data on fixed and mobile CBCT-guided bronchoscopy summarized by Wijma and coworkers [25]. To put this into perspective, the median total DAP (fluoroscopy and mCBCT spins) described in our trial (50.5 Gy-cm^2.^) is within range of the DAP associated with a CT of the chest (40–60 Gy-cm^2^) or CT of the abdomen and pelvis (60–80 Gy-cm^2^) [26]. Considering that we are obtaining tissue, which can provide diagnosis and profiling, the authors believe the radiation exposure associated with SS-RAB + mCBCT is certainly acceptable. Nevertheless, we cannot overemphasize the need to follow the ALARA principle (as low as reasonably achievable), and we should also focus on limiting radiation exposure associated with 2D fluoroscopy, which, in our study, was greater than that associated with mCBCT spins.

Our current study has a few limitations, such as being a single-center study, which can reduce the generalizability of its results. Having said so, multiple (five) operators, with differing degrees of experience at the time of this study, performed these procedures. Another critique of the study could be the fact that RP-EBUS was not utilized to confirm target reach after robotic navigation. This, and the fact that we did not take random blind biopsies around the virtual target (“4-quadrant” or “cloud” biopsies), may be the reasons for such a low TIL seen at the first mCBCT. Our trial cannot inform the readers whether the combination of SS-RAB + mCBCT is superior to the combination of SS-RAB + RP-EBUS. A prospective multicenter trial with SS-RAB + RP-EBUS (PRECISE trial, NCT03893539) has been conducted and we hope its results will be published soon. Nevertheless, comparative studies would be needed to make this assertion. In practices where CBCT is not available, RP-EBUS is still of paramount importance, since it is the only method to confirm that the target has been reached. We did not utilize RP-EBUS in our study because, in the authors’ opinion, when CBCT is available and being utilized to confirm TIL, the use of RP-EBUS to confirm target reach is no longer necessary. RP-EBUS cannot confirm TIL and regardless of the RP-EBUS image obtained, a CBCT is needed when TIL confirmation is desired. Moreover, concentric RP-EBUS images—the best-case scenario—do not guarantee TIL since biopsy tools are stiffer than radial probes and can follow a different trajectory, and RP-EBUS images may be falsely positive due to atelectasis. Furthermore, eccentric RP-EBUS images are even less helpful since they cannot accurately tell us on which side of the probe the target is located. Finally, most of the targets now sampled with CBCT guidance do not have a bronchus sign (an airway that is in contact with the lesion), and such, they cannot even be seen with RP-EBUS. With regards to taking “4-quadrant” or “cloud” biopsies to increase the chances of hitting a target, it is a practice that we strongly discourage when CBCT imaging is available. There is no reason to “blindly” sample normal parenchyma (potentially hitting vessels or pleura), thus increasing the length of the procedure, the associated risks, the number of samples to be read by pathologists, and the overall cost, when 3D imaging is available.

Not a true limitation, but rather an important clarification, is that the newer software version of SS-RAB that integrates the mCBCT images obtained intraoperatively and updates the location of the virtual target was not available at the time this trial was designed. Thus, all adjustments after obtaining mCBCT spins were performed based on the proceduralists’ interpretation of the mCBCT images. It is hypothesized that the “integration” may help reduce human error in these fine adjustments needed to obtain TIL, with a potential decrease in the number of spins to achieve TIL, and maybe also improved outcomes. A multicenter prospective trial of “integrated” SS-RAB + mCBCT has finalized enrollment (NCT05562895) and its results will soon be available.

Lastly, we would like to comment on the smaller but still significant gap between the rate of TIL and the diagnostic yield. While the exact explanation for this gap is not clear, we believe that multiple factors can influence our ability to obtain diagnostic material in these cases: the specific histology, the type of tumor (solid vs. non-solid), the vascularity of the tumor, and the type of tool we utilize to sample the tumor. We remain hopeful that better tools will allow us to close this gap further in the near future.

## 5. Conclusions

The addition of mCBCT guidance to SS-RAB allows bronchoscopists to compensate for CTBD, leading to an increase in the rate of TIL and diagnostic yield. The combination of these two technologies also has the potential benefit of reducing the rate of pneumothorax in peripheral bronchoscopy, though this finding needs to be further corroborated in larger clinical trials. The radiation exposure associated with the use of fluoroscopy and mCBCT to guide SS-RAB is well within the acceptable ranges for a diagnostic procedure.

## Figures and Tables

**Figure 1 diagnostics-14-01955-f001:**
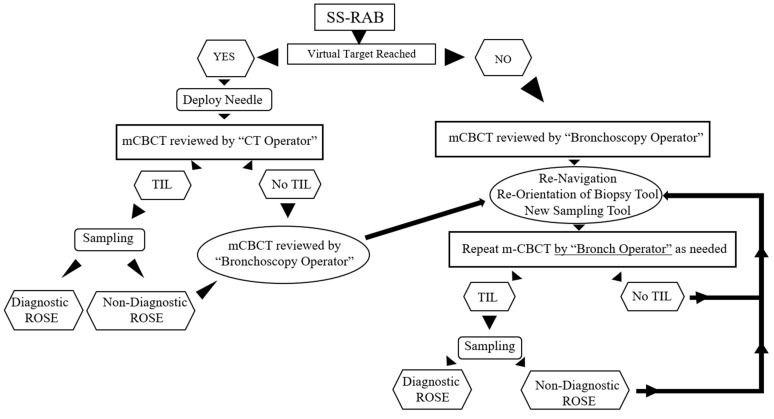
Study flowchart. mCBCT = mobile cone-beam CT; TIL = tool in lesion; ROSE = rapid on-site cytology examination.

**Table 1 diagnostics-14-01955-t001:** Baseline characteristics.

Characteristics	N = 67
Gender (%)	
- Female	37 (55)
- Male	30 (45)
Smoking History,	
- Never	22 (33)
- Ex-smoker	34 (51)
- Current	11 (16)
Prior Malignancy (*n* = 37)	
- Lung cancer	7 (19)
- Others	30 (81)
BMI	
Median (range)	27.9 (16.9–44.2)
ECOG	
Median (range)	1 (0–2)
ASA Score	
Median (range)	3 (1–3)
Target Size (cm)	
Median (range)	1.7 (0.9–3)
Target Characteristics	
- Solid	62 (93)
- Semi-solid	5 (7)
Target Location	
- Right Upper Lobe	22 (33)
- Right Middle Lobe	6 (9)
- Right Lower Lobe	13 (19)
- Left Upper Lobe	19 (29)
- Left Lower Lobe	7 (10)
Bronchus Sign	25 (37)
Distance to Pleura (cm)	
Median (range)	0.73 (0–4.7)
PET Avidity (*n* = 34)	29 (85)

BMI = Body Mass Index; ECOG = Eastern Cooperative Oncology Group; ASA = American Society of Anesthesiology; SD: standard deviation; PET = positron emission tomography.

**Table 2 diagnostics-14-01955-t002:** Procedure characteristics.

Procedure Characteristics	N = 67
Total Procedure Time (min)	
Mean (range)	51 (21–127)
Mediastinal Staging	47 (70)
Virtual Target Reached	65 (97)
Target Visualized by 2D Fluoroscopy	19 (28)
Sampling Tools (%)	
- Needle	67 (100)
- Cytology brush	25 (37)
- Forceps or Cryobiopsy	24 (36)
First mCBCT Spin Findings	
- No TIL	44 (66)
- TIL/center of lesion	18 (27)
- TIL/periphery of lesion	5 (7)
% of Patients with TIL at Each mCBCT Spin	
- 1st spin	34.3
- 2nd spin	58.3
- 3rd spin	70.4
- 4th spin	81.8
- 5th spin	98.6
Distance from Needle to Target (cm) (*n* = 44) *	
Median (interquartile range)	0.2 (0.47)
Fluoroscopy Time (min)	
Median (interquartile range)	8.5 (4.85)
Number of CBCT Spins	
Median (interquartile range)	3 (1.5)

CBCT = cone-beam computed tomography; TIL = tool in lesion; ***** calculated in 44 patients with no TIL at first mCBCT spin.

**Table 3 diagnostics-14-01955-t003:** Diagnosis obtained via bronchoscopy and post-bronchoscopy via other means.

Diagnosis Obtained via Bronchoscopy		Diagnosis Obtained “Post” Bronchoscopy *	
Malignant (*n* = 52) -Lung Adenocarcinoma -Lung Squamous Cell Carcinoma -Non-Small Cell Lung Cancer -Small Cell Lung Cancer -Carcinoid -Head and Neck Squamous Cell Carcinoma -Melanoma -Renal Cell Carcinoma	29 9 3 3 4 2 1 1	Malignant (*n* = 7) -Lung Squamous Cell carcinoma -Lung Adenocarcinoma -Colorectal carcinoma -Clear Cell Hemangioblastoma-Like Stromal Tumor of the Lung -Mantle cell lymphoma	2 2 1 1 1
Benign (*n* = 6) -Cryptococcal infection -Mucoid impaction due to bronchial atresia -Haemophilus influenzae pneumonia -Organizing pneumonia -Granulomatous inflammation -Amyloidoma	1 1 1 1 1 1	Benign (*n* = 2) -Acute inflammation in CT-FNA -No further samples	1 1
Non-Diagnostic Samples (*n* = 9) -Bronchial cells -Atypical cells -Acute inflammation -Atypical lymphocytes -Necrosis	2 2 2 1 2		

* This includes the 11 patients with non-diagnostic bronchoscopy who were later diagnosed via CT-TTNA or surgery, or whose lesions improved or resolved at follow up chest CT with no further samples.

**Table 4 diagnostics-14-01955-t004:** Fluoroscopy and mobile cone-beam CT-associated radiation exposure (*n* = 63).

	Total Fluoroscopy Time	Number of mCBCT Spins	Total Exposure (Entire Procedure)	Reference Air Kerma (Entire Procedure)	Total Exposure from mCBCT Spins	Reference Air Kerma for Spins
	min		Gy-cm^2^	mGy	Gy-cm^2^	mGy
**Median**	8.5	3	50.55	363.2	19.83	124.9
**IQR**	4.85	1.5	34.50	249.1	17.08	107.55

## Data Availability

The data are contained within the article.

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
