# Peer review of "Improving Shape-Sensing Robotic-Assisted Bronchoscopy Outcomes with Mobile Cone-Beam Computed Tomography Guidance"

_diagnostics, 2024, doi:10.3390/diagnostics14171955_

Round 1
Reviewer 1 Report
Comments and Suggestions for Authors
Dear authors,
My compliments for a very worthwhile manuscript that adds to the perspective of the field. Studying the value of ssRAB alone and in combination with mCBCT gives a valuable perspective in the optimization of the yield for pulmonary lesion diagnostics.
Please find all my minor comments/suggestions in the PDF attached.
One last question I would like to add on top of the remarks/questions in the PDF: Do you think your findings would have been significantly different if you had fCBCT at your disposal instead of mCBCT?

Two small spelling mistakes were detected, see attached file for highlights.
Author Response
Comment 1:
while the majority of readers will know the topic, might be good to introduce that a mobile CBCT is basically a C-arm with more advanced soft- and hardware that has become available in recent years due to technological development.
Response 1: Thanks for the input. We have entered a short sentence introducing mobile CBCT.
Comment 2:
Can a version be supplied such that it is 'specific' point of software?
Response 2: yes, the integrated version is called P4. This was added. Thanks.
Comment 3:
Can a version be supplied such that it is 'specific' point of software?
Response 3: Thanks, this was added.
Comment 4: From own experience; pre-op CT evaluation will not always visualize endobronchial pathways. What did you do for potential trans-parenchymal access cases? Please elaborate.
Response 4: Thanks for your comment. The majority of the cases were transparenchymal access in this study and in our practice. Fortunately, we do not need airways leading to the lesion with this technique. We navigate to the closest airway, aim at the virtual target, deploy needle (through bronchial wall and lung) and then scan.
Comment 5: If not later on discussed in discussion section: Was there any bias of the operator because he/she knew subsequent imaging would be available if unsuccessful? -> I will keep the needle a little bit proximal, because the lesion is at the pleura, and I will receive imaging information if no TIL is obtained anyhow?
Response 5: Thanks for your comment. We tried to avoid this potential bias, and the operator had to follow the protocol and insert the needle halfway between the "near" and "far" ends of the target, to aim for the center.
Comment 6:
was only needle used for obtaining samples?
Response 6:. Thanks for your comment. The answer is no, as described in table 2, other tools were utilized. The bare minimum per protocol was 3 TBNAs (since we needed to obtain ROSE). We better explained this in methods now. Thanks.
Comment 7:
see earlier; if only 37.3%, how did you conclude that needle positioned was correct before going towards mCBCT imaging? Purely on VBN + fluoroscopy?
Response:
The operator had to estimate the length of the needle that was needed to reach the target based on the near and far distances and aim towards the target center. The conclusion that the needle was in the target was exclusively after analyzing mCBCT. The operator had to trust the virtual navigation of the SS-RAB and deploy the needle on the virtual target, not knowing if it would be on the actual target or not. That was the purpose of the study (if we hit the virtual target how often are we on the actual target, and how much can we improve that after analyzing the mCBCT images). Thanks.
Comment 8:
what is the procedure time definition? This is heterogeneously used in literature, please elaborate in methods section.
Thank you so much. The definition of procedure time was already in lines 156-157 of the methods section, under the subheading of "Primary, Secondary Endpoints and Definitions".
Comment 9:
I do not fully understand the meaning of this sentence; distance to outer edge or center? Distance of needle to target in cases where no TIL was achieved by ssRAB alone? Even though this is also further discussed in discussion, might be good to repeat here for clarification.
Response 9: Thanks. The method we utilized to measure this distance was explained in methods section:
..."the shortest distance from the needle to the target (“tip” of the needle to “edge” of the target) observed in the first mCBCT spin was recorded for patients who did not have TIL (in whichever axis this shortest distance was found)"....
Comment 10:
The amount of exposure caused by fluoroscopy tops the amount of exposure caused by CBCT. This is relatively unexpected, considering a median of 3 CBCTs was made. Could you please comment on the use of collimation or other dose limiting efforts during fluoroscopy (ie aspects which are also mentioned in the WABIP position paper you mentioned) -> could you have lowered the dose?
Response 10: Excellent point. We have in fact noticed this and talked among ourselves about possible ways to address this and lower the radiation associated with fluoroscopy. We have added the following to the discussion:
"Nevertheless, we cannot overemphasize the need to follow the ALARA principle (as low as reasonably achievable), and we will focus on limiting radiation exposure associated with fluoroscopy, which, in our study, was greater than that associated with mCBCT spins.".
Comment 11:
I would not put this forward as factual, but rather as an opinion. In our practice, where we routinely use RP-EBUS as well as fCBCT as well as ROSE, the imaging as obtained with RP-EBUS can give us very fast information on needed adjustment of angulation with ssRAB. I'm safe to say it can give us information that would allow us to circumvent using CBCT; if central TIL is achieved with RPEBUS, CBCT is unnecessary. The use of this technology also has important consequences for the amount of radiation used.
Response 11:
Thanks for your comment. We have "toned down" this comment, which, as you rightfully state, it is only our opinion. Thanks again.
Comment 12:
there is a feature on the ssRAB (ION) machine that does help you in these cases when also having rEBUS.
> biopsy tab > after rEBUS input you can select a positioning guide.
After doing so, you'll get a virtual 'ring' around the lesion on your top screen which allows you to angulate your ssRAB with rEBUS in. This'll help you optimize your angulation such that it might have better 'contact' with your lesion (as can then be seen on the rEBUS image).
If trans-parenchymal access is needed it will be of little help (to my experience), but in other cases it might be quite helpful.
Concluding: your point is not complete, please reconsider.
Response 12: Thanks. We have not found this method to be consistently accurate when we were using RP-EBUS within the prior trial, the PRECISE trial.
Comment 13:
question: if ROSE was available, why did you stop sampling if i.e. only normal bronchial cells were found? Could you please elaborate on your idea why the TIL versus diagnostic yield is still >10% even in CBCT confirmed positioning? How can we close the gap?
Response 13:
Great point, and this is the million dollar question, how to close the GAP from TIL to diagnosis, which is in the 10-15% range. We have added the following to the discussion:
"Lastly, we would like to comment on the smaller, but still significant gap between the rate of TIL and the diagnostic yield. While the exact explanation for this gap is not clear, we believe that multiple factors can influence our ability to obtain diagnostic material in these cases: the specific histology, the type of tumor (solid vs. non-solid), the vascularity of the tumor, and the type of tool we utilize to sample the tumor. We remain hopeful that better tools will allow us to close this gap further in the near future."
Comment 14:
Do you think your findings would have been significantly different if you had fCBCT at your disposal instead of mCBCT?
Response 14: Thanks. We do not think so. The only case in which we could not achieve tool in lesion, it was due to the exit angle of the needle which could not exit the robotic catheter. But we otherwise had a clear view of the target and tool in 100% of the cases. A better-quality image would have not influenced our results.
Reviewer 2 Report
Comments and Suggestions for Authors
This manuscript is a prospective single-center study on SS-RAB with mCBCT for peripheral pulmonary lesions. It is an excellent study demonstrating that the addition of mCBCT improves the TIL rate and diagnostic yield. My comments are as follows:
Major comments
Lines 307-321.
The section addressing RP-EBUS lacks supporting references or data analysis. Consider removing redundant explanations and speculative statements.
Minor comments
Table 1
Spell out "BMI" and "PET."
Table 2
Please specify the histological types for "Head and Neck."
Amyloidoma and Hemangioblastoma are benign conditions.
Can bronchial atresia be pathologically confirmed with micro-specimens obtained via bronchoscopy? Isn't this non-diagnostic?
There are multiple instances where spaces are missing before "cm."
Author Response
Comment 1:
Lines 307-321.
The section addressing RP-EBUS lacks supporting references or data analysis. Consider removing redundant explanations and speculative statements.
Response:
Thanks for your comment. We have clarified at the beginning of this section that this is the opinion of the authors, not facts. We do believe an explanation on why we did not utilize RP-EBUS is relevant since most people still do. Thanks again.
Comment 2:
Table 1
Spell out "BMI" and "PET."
Response 2:
Thanks. This has been spelled out.
Comment 3:
Table 2
Please specify the histological types for "Head and Neck."
Amyloidoma and Hemangioblastoma are benign conditions.
Can bronchial atresia be pathologically confirmed with micro-specimens obtained via bronchoscopy? Isn't this non-diagnostic?
Response 2:
Thanks for highlighting this. We have specified that it was a Head and Neck squamous cell carcinoma.
You are correct that the Amyloidoma is a benign tumor, it was accidentally placed under malignant conditions because it was due to Multiple Myeloma. We have corrected this.
The Hemangioblastoma was actually a Clear Cell Hemangioblastoma-Like Stromal Tumor of the Lung, a very rare solid malignant tumor which was treated with chemotherapy. Thanks for catching this error. It will stay under malignant with the proper terminology.
With regards to the bronchial atresia and mucoid impaction, what presented like a lung nodule was the mucoid impaction. The diagnosis was a combination of the cytology (aspiration of mucin), the finding on camera of bronchial atresia (we pierced through this with the needle), and the fact that after we aspirated the mucin the lesion collapsed and almost disappeared in close follow up. We believe this bronchoscopy was diagnostic. We re-phrased this as Mucoid impaction due to bronchial atresia, since mucin is what the cytologists found.
Comment 3:
There are multiple instances where spaces are missing before "cm."
Response 3:. Thanks, we have corrected these.
Round 2
Reviewer 2 Report
Comments and Suggestions for Authors
Thank you for your thoughtful and sincere response to my comments. I have just one final comment.
If CBCT is available, I can fully understand and agree with the authors' perspective. However, if CBCT is not available, I believe there is merit in using RP-EBUS. As evidence, the TIL rate and diagnostic yield of SS-RAB alone in this study, without RP-EBUS, were lower than those reported in previous studies. Including this point would contribute to a more balanced discussion.
Author Response
Comment 1:
If CBCT is available, I can fully understand and agree with the authors' perspective. However, if CBCT is not available, I believe there is merit in using RP-EBUS. As evidence, the TIL rate and diagnostic yield of SS-RAB alone in this study, without RP-EBUS, were lower than those reported in previous studies. Including this point would contribute to a more balanced discussion.
Response 1:
Thanks for your comment. We agree with your point and have added the following to the manuscript's discussion.
....."In practices where CBCT is not available, RP-EBUS is still of paramount importance, since it is the only method to confirm that target has been reached. We did not utilize RP-EBUS in our study because, in the authors’ opinion, when CBCT is available and being utilized to confirm TIL, a second and less accurate confirmation of target reach with RP-EBUS is not necessary....".